# Non-Convex Finite-Sum Optimization
# Via SCSG Methods

**Lihua Lei**
UC Berkeley
lihua.lei@berkeley.edu

**Cheng Ju**
UC Berkeley
cju@berkeley.edu

**Jianbo Chen**
UC Berkeley
jianbochen@berkeley.edu

**Michael I. Jordan**
UC Berkeley
jordan@stat.berkeley.edu

## Abstract

We develop a class of algorithms, as variants of the *stochastically controlled stochastic gradient* (SCSG) methods [21], for the smooth non-convex finite-sum optimization problem. Assuming the smoothness of each component, the complexity of SCSG to reach a stationary point with $\mathbb{E}\|\nabla f(x)\|^2 \leq \varepsilon$ is $O\left(\min\{\varepsilon^{-5/3}, \varepsilon^{-1}n^{2/3}\}\right)$, which strictly outperforms the stochastic gradient descent. Moreover, SCSG is never worse than the state-of-the-art methods based on variance reduction and it significantly outperforms them when the target accuracy is low. A similar acceleration is also achieved when the functions satisfy the Polyak-Lojasiewicz condition. Empirical experiments demonstrate that SCSG outperforms stochastic gradient methods on training multi-layers neural networks in terms of both training and validation loss.

## 1 Introduction

We study smooth non-convex finite-sum optimization problems of the form

$$\min_{x \in \mathbb{R}^d} f(x) = \frac{1}{n} \sum_{i=1}^{n} f_i(x) \tag{1}$$

where each component $f_i(x)$ is possibly non-convex with Lipschitz gradients. This generic form captures numerous statistical learning problems, ranging from generalized linear models [22] to deep neural networks [19].

In contrast to the convex case, the non-convex case is comparatively under-studied. Early work focused on the asymptotic performance of algorithms [11, 7, 29], with non-asymptotic complexity bounds emerging more recently [24]. In recent years, complexity results have been derived for both gradient methods [13, 2, 8, 9] and stochastic gradient methods [12, 13, 6, 4, 26, 27, 3]. Unlike in the convex case, in the non-convex case one can not expect a gradient-based algorithm to converge to the global minimum if only smoothness is assumed. As a consequence, instead of measuring function-value suboptimality $\mathbb{E}f(x) - \inf_x f(x)$ as in the convex case, convergence is generally measured in terms of the squared norm of the gradient; i.e., $\mathbb{E}\|\nabla f(x)\|^2$. We summarize the best available rates [1] in Table 1. We also list the rates for Polyak-Lojasiewicz (P-L) functions, which will be defined in Section 2. The accuracy for minimizing P-L functions is measured by $\mathbb{E}f(x) - \inf_x f(x)$.

Table 1: Computation complexity of gradient methods and stochastic gradient methods for the finite-sum non-convex optimization problem (1). The second and third columns summarize the rates in the smooth and P-L cases respectively. $\mu$ is the P-L constant and $\mathcal{H}^*$ is the variance of a stochastic gradient. These quantities are defined in Section 2. The final column gives additional required assumptions beyond smoothness or the P-L condition. The symbol $\wedge$ denotes a minimum and $\tilde{O}(\cdot)$ is the usual Landau big-O notation with logarithmic terms hidden.

| | Smooth | Polyak-Lojasiewicz | additional cond. |
|---|---|---|---|
| **Gradient Methods** | | | |
| GD | $O\left(\frac{n}{\varepsilon}\right)$ [24, 13] | $\tilde{O}\left(\frac{n}{\mu}\right)$ [25, 17] | - |
| Best available | $\tilde{O}\left(\frac{n}{\varepsilon^{7/8}}\right)$ [9] | - | smooth gradient |
| | $\tilde{O}\left(\frac{n}{\varepsilon^{5/6}}\right)$ [9] | - | smooth Hessian |
| **Stochastic Gradient Methods** | | | |
| SGD | $O\left(\frac{1}{\varepsilon^2}\right)$ [24, 26] | $O\left(\frac{1}{\mu^2\varepsilon}\right)$ [17] | $\mathcal{H}^* = O(1)$ |
| Best available | $O\left(n + \frac{n^{2/3}}{\varepsilon}\right)$ [26, 27] | $\tilde{O}\left(n + \frac{n^{2/3}}{\mu}\right)$ [26, 27] | - |
| SCSG | $\tilde{O}\left(\frac{1}{\varepsilon^{5/3}} \wedge \frac{n^{2/3}}{\varepsilon}\right)$ | $\tilde{O}\left((\frac{1}{\mu\varepsilon} \wedge n) + \frac{1}{\mu}(\frac{1}{\mu\varepsilon} \wedge n)^{2/3}\right)$ | $\mathcal{H}^* = O(1)$ |

As in the convex case, gradient methods have better dependence on $\varepsilon$ in the non-convex case but worse dependence on $n$. This is due to the requirement of computing a full gradient. Comparing the complexity of SGD and the best achievable rate for stochastic gradient methods, achieved via variance-reduction methods, the dependence on $\varepsilon$ is significantly improved in the latter case. However, unless $\varepsilon \ll n^{-1/2}$, SGD has similar or even better theoretical complexity than gradient methods and existing variance-reduction methods. In practice, it is often the case that $n$ is very large ($10^5 \sim 10^9$) while the target accuracy is moderate ($10^{-1} \sim 10^{-3}$). In this case, SGD has a meaningful advantage over other methods, deriving from the fact that it does not require a full gradient computation. This motivates the following research question: Is there an algorithm that

- achieves/beats the theoretical complexity of SGD in the regime of modest target accuracy;

- and achieves/beats the theoretical complexity of existing variance-reduction methods in the regime of high target accuracy?

The question has been partially answered in the convex case by [21] in their formulation of the *stochastically controlled stochastic gradient* (SCSG) methods. When the target accuracy is low, SCSG has the same $O\left(\varepsilon^{-2}\right)$ rate as SGD but with a much smaller data-dependent constant factor (which does not even require bounded gradients). When the target accuracy is high, SCSG achieves the same rate as the best non-accelerated methods, $O(\frac{n}{\varepsilon})$. Despite the gap between this and the optimal rate, SCSG is the first known algorithm that provably achieves the desired performance in both regimes.

In this paper, we generalize SCSG to the non-convex setting which, surprisingly, provides a completely affirmative answer to the question raised before. By only assuming smoothness of each component as in almost all other works, SCSG is always $O\left(\varepsilon^{-1/3}\right)$ faster than SGD and is never worse than recently developed stochastic gradient methods that achieve the best rate. When $\varepsilon \gg \frac{1}{n}$, SCSG is at least $O((\varepsilon n)^{2/3})$ faster than the best SVRG-type algorithms. Comparing with the gradient methods, SCSG has a better convergence rate provided $\varepsilon \gg n^{-6/5}$, which is the common setting in practice. Interestingly, there is a parallel to recent advances in gradient methods; [9] improved the classical $O(\varepsilon^{-1})$ rate of gradient descent to $O(\varepsilon^{-5/6})$; this parallels the improvement of SCSG over SGD from $O(\varepsilon^{-2})$ to $O(\varepsilon^{-5/3})$.

Beyond the theoretical advantages of SCSG, we also show that SCSG yields good empirical performance for the training of multi-layer neural networks. It is worth emphasizing that the mechanism by which SCSG achieves acceleration (variance reduction) is qualitatively different from other speed-up

techniques, including momentum [28] and adaptive stepsizes [18]. It will be of interest in future work to explore combinations of these various approaches in the training of deep neural networks.

The rest of paper is organized as follows: In Section 2 we discuss our notation and assumptions and we state the basic SCSG algorithm. We present the theoretical convergence analysis in Section 3. Experimental results are presented in Section 4. All the technical proofs are relegated to the Appendices. Our code is available at `https://github.com/Jianbo-Lab/SCSG`.

## 2 Notation, Assumptions and Algorithm

We use $\|\cdot\|$ to denote the Euclidean norm and write $\min\{a, b\}$ as $a \wedge b$ for brevity throughout the paper. The notation $\tilde{O}$, which hides logarithmic terms, will only be used to maximize readability in our presentation but will not be used in the formal analysis.

We define computation cost using the IFO framework of [1] which assumes that sampling an index $i$ and accessing the pair $(\nabla f_i(x), f_i(x))$ incur a unit of cost. For brevity, we write $\nabla f_{\mathcal{I}}(x)$ for $\frac{1}{|\mathcal{I}|} \sum_{i \in \mathcal{I}} \nabla f_i(x)$. Note that calculating $\nabla f_{\mathcal{I}}(x)$ incurs $|\mathcal{I}|$ units of computational cost. $x$ is called an $\varepsilon$-accurate solution iff $\mathbb{E}\|\nabla f(x)\|^2 \leq \varepsilon$. The minimum IFO complexity to reach an $\varepsilon$-accurate solution is denoted by $C_{\text{comp}}(\varepsilon)$.

Recall that a random variable $N$ has a geometric distribution, $N \sim \text{Geom}(\gamma)$, if $N$ is supported on the non-negative integers [2] with

$$P(N = k) = \gamma^k(1 - \gamma), \quad \forall k = 0, 1, \dots$$

An elementary calculation shows that

$$\mathbb{E}_{N \sim \text{Geom}(\gamma)} = \frac{\gamma}{1 - \gamma}. \tag{2}$$

To formulate our complexity bounds, we define

$$f^* = \inf_x f(x), \quad \Delta_f = f(\tilde{x}_0) - f^*.$$

Further we define $\mathcal{H}^*$ as an upper bound of the variance of stochastic gradients, i.e.

$$\mathcal{H}^* = \sup_x \frac{1}{n} \sum_{i=1}^n \|\nabla f_i(x) - \nabla f(x)\|^2. \tag{3}$$

The assumption **A1** on the smoothness of individual functions will be made throughout this paper.

**A1** $f_i$ is differentiable with
$$\|\nabla f_i(x) - \nabla f_i(y)\| \leq L\|x - y\|$$
for some $L < \infty$ and all $i \in \{1, \dots, n\}$.

As a direct consequence of assumption **A1**, it holds for any $x, y \in \mathbb{R}^d$ that

$$-\frac{L}{2}\|x - y\|^2 \leq f_i(x) - f_i(y) - \langle \nabla f_i(y), x - y \rangle \leq \frac{L}{2}\|x - y\|^2. \tag{4}$$

In this paper, we also consider the following Polyak-Lojasiewicz (PL) condition [25]. It is weaker than strong convexity as well as other popular conditions that appeared in optimization literature; see [17] for an extensive discussion.

**A2** $f(x)$ satisfies the P-L condition with $\mu > 0$ if
$$\|\nabla f(x)\|^2 \geq 2\mu(f(x) - f(x^*))$$
where $x^*$ is the global minimum of $f$.

## 2.1 Generic form of SCSG methods

The algorithm we propose in this paper is similar to that of [14] except (critically) the number of inner loops is a geometric random variable. This is an essential component in the analysis of SCSG, and, as we will show below, it is key in allowing us to extend the complexity analysis for SCSG to the non-convex case. Moreover, that algorithm that we present here employs a mini-batch procedure in the inner loop and outputs a random sample instead of an average of the iterates. The pseudo-code is shown in Algorithm 1.

---

**Algorithm 1** (Mini-Batch) Stochastically Controlled Stochastic Gradient (SCSG) method for smooth non-convex finite-sum objectives

---

**Inputs:** Number of stages $T$, initial iterate $\tilde{x}_0$, stepsizes $(\eta_j)_{j=1}^T$, batch sizes $(B_j)_{j=1}^T$, mini-batch sizes $(b_j)_{j=1}^T$.

**Procedure**

  1: **for** $j = 1, 2, \cdots, T$ **do**
  2:     Uniformly sample a batch $\mathcal{I}_j \subset \{1, \cdots, n\}$ with $|\mathcal{I}_j| = B_j$;
  3:     $g_j \leftarrow \nabla f_{\mathcal{I}_j}(\tilde{x}_{j-1})$;
  4:     $x_0^{(j)} \leftarrow \tilde{x}_{j-1}$;
  5:     Generate $N_j \sim \mathrm{Geom}\,(B_j/(B_j + b_j))$;
  6:     **for** $k = 1, 2, \cdots, N_j$ **do**
  7:          Randomly pick $\tilde{\mathcal{I}}_{k-1} \subset [n]$ with $|\tilde{\mathcal{I}}_{k-1}| = b_j$;
  8:          $\nu_{k-1}^{(j)} \leftarrow \nabla f_{\tilde{\mathcal{I}}_{k-1}}(x_{k-1}^{(j)}) - \nabla f_{\tilde{\mathcal{I}}_{k-1}}(x_0^{(j)}) + g_j$;
  9:          $x_k^{(j)} \leftarrow x_{k-1}^{(j)} - \eta_j \nu_{k-1}^{(j)}$;
10:     **end for**
11:     $\tilde{x}_j \leftarrow x_{N_j}^{(j)}$;
12: **end for**

**Output:** (Smooth case) Sample $\tilde{x}_T^*$ from $(\tilde{x}_j)_{j=1}^T$ with $P(\tilde{x}_T^* = \tilde{x}_j) \propto \eta_j B_j / b_j$; (P-L case) $\tilde{x}_T$.

---

As seen in the pseudo-code, the SCSG method consists of multiple epochs. In the $j$-th epoch, a mini-batch of size $B_j$ is drawn uniformly from the data and a sequence of mini-batch SVRG-type updates are implemented, with the total number of updates being randomly generated from a geometric distribution, with mean equal to the batch size. Finally it outputs a random sample from $\{\tilde{x}_j\}_{j=1}^T$. This is the standard way, proposed by [23], as opposed to computing $\arg\min_{j \le T} \|\nabla f(\tilde{x}_j)\|$ which requires additional overhead. By (2), the average total cost is

$$\sum_{j=1}^T (B_j + b_j \cdot \mathbb{E}N_j) = \sum_{i=1}^T (B_j + b_j \cdot \frac{B_j}{b_j}) = 2\sum_{j=1}^T B_j. \tag{5}$$

Define $T(\varepsilon)$ as the minimum number of epochs such that all outputs afterwards are $\varepsilon$-accurate solutions, i.e.

$$T(\varepsilon) = \min\{T : \mathbb{E}\|\nabla f(\tilde{x}_{T'}^*)\| \le \varepsilon \text{ for all } T' \ge T\}.$$

Recall the definition of $C_{\mathrm{comp}}(\varepsilon)$ at the beginning of this section, the average IFO complexity to reach an $\varepsilon$-accurate solution is

$$\mathbb{E}C_{\mathrm{comp}}(\varepsilon) \le 2\sum_{j=1}^{T(\varepsilon)} B_j.$$

## 2.2 Parameter settings

The generic form (Algorithm 1) allows for flexibility in both stepsize, $\eta_j$, and batch/mini-batch size, $(B_j, b_j)$. In order to minimize the amount of tuning needed in practice, we provide several default settings which have theoretical support. The settings and the corresponding complexity results are summarized in Table 2. Note that all settings fix $b_j = 1$ since this yields the best rate as will be shown in Section 3. However, in practice a reasonably large mini-batch size $b_j$ might be favorable due to the acceleration that could be achieved by vectorization; see Section 4 for more discussions on this point.

Table 2: Parameter settings analyzed in this paper.

| | $\eta_j$ | $B_j$ | $b_j$ | Type of Objectives | $\mathbb{E}C_{\mathrm{comp}}(\varepsilon)$ |
|---|---|---|---|---|---|
| Version 1 | $\frac{1}{2LB^{2/3}}$ | $O\left(\frac{1}{\varepsilon} \wedge n\right)$ | 1 | Smooth | $O\left(\frac{1}{\varepsilon^{5/3}} \wedge \frac{n^{2/3}}{\varepsilon}\right)$ |
| Version 2 | $\frac{1}{2LB_j^{2/3}}$ | $j^{\frac{3}{2}} \wedge n$ | 1 | Smooth | $\tilde{O}\left(\frac{1}{\varepsilon^{5/3}} \wedge \frac{n^{2/3}}{\varepsilon}\right)$ |
| Version 3 | $\frac{1}{2LB_j^{2/3}}$ | $O\left(\frac{1}{\mu\varepsilon} \wedge n\right)$ | 1 | Polyak-Lojasiewicz | $\tilde{O}\left((\frac{1}{\mu\varepsilon} \wedge n) + \frac{1}{\mu}(\frac{1}{\mu\varepsilon} \wedge n)^{2/3}\right)$ |

# 3 Convergence Analysis

## 3.1 One-epoch analysis

First we present the analysis for a single epoch. Given $j$, we define

$$e_j = \nabla f_{\mathcal{I}_j}(\tilde{x}_{j-1}) - \nabla f(\tilde{x}_{j-1}). \tag{6}$$

As shown in [14], the gradient update $\nu_k^{(j)}$ is a biased estimate of the gradient $\nabla f(x_k^{(j)})$ conditioning on the current random index $i_k$. Specifically, within the $j$-th epoch,

$$\mathbb{E}_{\tilde{\mathcal{I}}_k} \nu_k^{(j)} = \nabla f(x_k^{(j)}) + \nabla f_{\mathcal{I}_j}(x_0^{(j)}) - \nabla f(x_0^{(j)}) = \nabla f(x_k^{(j)}) + e_j.$$

This reveals the basic qualitative difference between SVRG and SCSG. Most of the novelty in our analysis lies in dealing with the extra term $e_j$. Unlike [14], we do not assume $\|x_k^{(j)} - x^*\|$ to be bounded since this is invalid in unconstrained problems, even in convex cases.

By careful analysis of primal and dual gaps [cf. 5], we find that the stepsize $\eta_j$ should scale as $(B_j/b_j)^{-\frac{2}{3}}$. Then same phenomenon has also been observed in [26, 27, 4] when $b_j = 1$ and $B_j = n$.

**Theorem 3.1** *Let $\eta_j L = \gamma(B_j/b_j)^{-\frac{2}{3}}$. Suppose $\gamma \leq \frac{1}{3}$ and $B_j \geq 8b_j$ for all $j$, then under Assumption A1,*

$$\mathbb{E}\|\nabla f(\tilde{x}_j)\|^2 \leq \frac{5L}{\gamma} \cdot \left(\frac{b_j}{B_j}\right)^{\frac{1}{3}} \mathbb{E}(f(\tilde{x}_{j-1}) - f(\tilde{x}_j)) + \frac{6I(B_j < n)}{B_j} \cdot \mathcal{H}^*. \tag{7}$$

The proof is presented in Appendix B. It is not surprising that a large mini-batch size will increase the theoretical complexity as in the analysis of mini-batch SGD. For this reason we restrict most of our subsequent analysis to $b_j \equiv 1$.

## 3.2 Convergence analysis for smooth non-convex objectives

When only assuming smoothness, the output $\tilde{x}_T^*$ is a random element from $(\tilde{x}_j)_{j=1}^T$. Telescoping (7) over all epochs, we easily obtain the following result.

**Theorem 3.2** *Under the specifications of Theorem 3.1 and Assumption A1,*

$$\mathbb{E}\|\nabla f(\tilde{x}_T^*)\|^2 \leq \frac{\frac{5L}{\gamma}\Delta_f + 6\left(\sum_{j=1}^T b_j^{-\frac{1}{3}} B_j^{-\frac{2}{3}} I(B_j < n)\right)\mathcal{H}^*}{\sum_{j=1}^T b_j^{-\frac{1}{3}} B_j^{\frac{1}{3}}}.$$

This theorem covers many existing results. When $B_j = n$ and $b_j = 1$, Theorem 3.2 implies that $\mathbb{E}\|\nabla f(\tilde{x}_T^*)\|^2 = O\left(\frac{L\Delta_f}{Tn^{1/3}}\right)$ and hence $T(\varepsilon) = O(1 + \frac{L\Delta_f}{\varepsilon n^{1/3}})$. This yields the same complexity bound $\mathbb{E}C_{\mathrm{comp}}(\varepsilon) = O(n + \frac{n^{2/3}L\Delta_f}{\varepsilon})$ as SVRG [26]. On the other hand, when $b_j = B_j \equiv B$ for some $B < n$, Theorem 3.2 implies that $\mathbb{E}\|\nabla f(\tilde{x}_T^*)\|^2 = O\left(\frac{L\Delta_f}{T} + \frac{\mathcal{H}^*}{B}\right)$. The second term can be made $O(\varepsilon)$ by setting $B = O\left(\frac{\mathcal{H}^*}{\varepsilon}\right)$. Under this setting $T(\varepsilon) = O\left(\frac{L\Delta_f}{\varepsilon}\right)$ and $\mathbb{E}C_{\mathrm{comp}}(\varepsilon) = O\left(\frac{L\Delta_f\mathcal{H}^*}{\varepsilon^2}\right)$. This is the same rate as in [26] for SGD.

However, both of the above settings are suboptimal since they either set the batch sizes $B_j$ too large or set the mini-batch sizes $b_j$ too large. By Theorem 3.2, SCSG can be regarded as an interpolation between SGD and SVRG. By leveraging these two parameters, SCSG is able to outperform both methods.

We start from considering a constant batch/mini-batch size $B_j \equiv B, b_j \equiv 1$. Similar to SGD and SCSG, $B$ should be at least $O(\frac{\mathcal{H}^*}{\varepsilon})$. In applications like the training of neural networks, the required accuracy is moderate and hence a small batch size suffices. This is particularly important since the gradient can be computed without communication overhead, which is the bottleneck of SVRG-type algorithms. As shown in Corollary 3.3 below, the complexity of SCSG beats both SGD and SVRG.

**Corollary 3.3** *(Constant batch sizes) Set*

$$b_j \equiv 1, \quad B_j \equiv B = \min\left\{\frac{12\mathcal{H}^*}{\varepsilon}, n\right\}, \quad \eta_j \equiv \eta = \frac{1}{6LB^{\frac{2}{3}}}.$$

*Then it holds that*

$$\mathbb{E}C_{\text{comp}}(\varepsilon) = O\left(\left(\frac{\mathcal{H}^*}{\varepsilon} \wedge n\right) + \frac{L\Delta_f}{\varepsilon} \cdot \left(\frac{\mathcal{H}^*}{\varepsilon} \wedge n\right)^{\frac{2}{3}}\right).$$

Assume that $L\Delta_f, \mathcal{H}^* = O(1)$, the above bound can be simplified to

$$\mathbb{E}C_{\text{comp}}(\varepsilon) = O\left(\left(\frac{1}{\varepsilon} \wedge n\right) + \frac{1}{\varepsilon} \cdot \left(\frac{1}{\varepsilon} \wedge n\right)^{\frac{2}{3}}\right) = O\left(\frac{1}{\varepsilon^{\frac{5}{3}}} \wedge \frac{n^{\frac{2}{3}}}{\varepsilon}\right).$$

When the target accuracy is high, one might consider a sequence of increasing batch sizes. Heuristically, a large batch is wasteful at the early stages when the iterates are inaccurate. Fixing the batch size to be $n$ as in SVRG is obviously suboptimal. Via an involved analysis, we find that $B_j \sim j^{\frac{3}{2}}$ gives the best complexity among the class of SCSG algorithms.

**Corollary 3.4** *(Time-varying batch sizes) Set*

$$b_j \equiv 1, \quad B_j = \min\left\{\lceil j^{\frac{3}{2}} \rceil, n\right\}, \quad \eta_j = \frac{1}{6LB_j^{\frac{2}{3}}}.$$

*Then it holds that*

$$\mathbb{E}C_{\text{comp}}(\varepsilon) = O\left(\min\left\{\frac{1}{\varepsilon^{\frac{5}{3}}}\left[(L\Delta_f)^{\frac{5}{3}} + (\mathcal{H}^*)^{\frac{5}{3}}\log^5\left(\frac{\mathcal{H}^*}{\varepsilon}\right)\right], n^{\frac{5}{3}}\right\} + \frac{n^{\frac{2}{3}}}{\varepsilon} \cdot (L\Delta_f + \mathcal{H}^*\log n)\right). \tag{8}$$

The proofs of both Corollary 3.3 and Corollary 3.4 are presented in Appendix C. To simplify the bound (8), we assume that $L\Delta_f, \mathcal{H}^* = O(1)$ in order to highlight the dependence on $\varepsilon$ and $n$. Then (8) can be simplified to

$$\mathbb{E}C_{\text{comp}}(\varepsilon) = O\left(\frac{1}{\varepsilon^{\frac{5}{3}}}\log^5\left(\frac{1}{\varepsilon}\right) \wedge n^{\frac{5}{3}} + \frac{n^{\frac{2}{3}}\log n}{\varepsilon}\right) = \tilde{O}\left(\frac{1}{\varepsilon^{\frac{5}{3}}} \wedge n^{\frac{5}{3}} + \frac{n^{\frac{2}{3}}}{\varepsilon}\right) = \tilde{O}\left(\frac{1}{\varepsilon^{\frac{5}{3}}} \wedge \frac{n^{\frac{2}{3}}}{\varepsilon}\right).$$

The log-factor $\log^5\left(\frac{1}{\varepsilon}\right)$ is purely an artifact of our proof. It can be reduced to $\log^{\frac{3}{2}+\mu}\left(\frac{1}{\varepsilon}\right)$ for any $\mu > 0$ by setting $B_j \sim j^{\frac{3}{2}}(\log j)^{\frac{3}{2}+\mu}$; see remark 1 in Appendix C.

## 3.3 Convergence analysis for P-L objectives

When the component $f_i(x)$ satisfies the P-L condition, it is known that the global minimum can be found efficiently by SGD [17] and SVRG-type algorithms [26, 4]. Similarly, SCSG can also achieve this. As in the last subsection, we start from a generic result to bound $\mathbb{E}(f(\tilde{x}_T) - f^*)$ and then consider specific settings of the parameters as well as their complexity bounds.

**Theorem 3.5** *Let $\lambda_j = \dfrac{5Lb_j^{\frac{1}{3}}}{\mu\gamma B_j^{\frac{1}{3}} + 5Lb_j^{\frac{1}{3}}}$. Then under the same settings of Theorem 3.2,*

$$\mathbb{E}(f(\tilde{x}_T) - f^*) \leq \lambda_T \lambda_{T-1} \ldots \lambda_1 \cdot \Delta_f + 6\gamma\mathcal{H}^* \cdot \sum_{j=1}^{T} \frac{\lambda_T \lambda_{T-1} \ldots \lambda_{j+1} \cdot I(B_j < n)}{\mu\gamma B_j + 5Lb_j^{\frac{1}{3}} B_j^{\frac{2}{3}}}.$$

The proofs and additional discussion are presented in Appendix D. Again, Theorem 3.5 covers existing complexity bounds for both SGD and SVRG. In fact, when $B_j = b_j \equiv B$ as in SGD, via some calculation, we obtain that

$$\mathbb{E}(f(\tilde{x}_T) - f^*) = O\left(\left(\frac{L}{\mu + L}\right)^T \cdot \Delta_f + \frac{\mathcal{H}^*}{\mu B}\right).$$

The second term can be made $O(\varepsilon)$ by setting $B = O(\frac{\mathcal{H}^*}{\mu\varepsilon})$, in which case $T(\varepsilon) = O(\frac{L}{\mu}\log\frac{\Delta_f}{\varepsilon})$. As a result, the average cost to reach an $\varepsilon$-accurate solution is $\mathbb{E}C_{\text{comp}}(\varepsilon) = O(\frac{L\mathcal{H}^*}{\mu^2\varepsilon})$, which is the same as [17]. On the other hand, when $B_j \equiv n$ and $b_j \equiv 1$ as in SVRG, Theorem 3.5 implies that

$$\mathbb{E}(f(\tilde{x}_T) - f^*) = O\left(\left(\frac{L}{\mu n^{\frac{1}{3}} + L}\right)^T \cdot \Delta_f\right).$$

This entails that $T(\varepsilon) = O\left((1 + \frac{1}{\mu n^{1/3}})\log\frac{1}{\varepsilon}\right)$ and hence $\mathbb{E}C_{\text{comp}}(\varepsilon) = O\left((n + \frac{n^{2/3}}{\mu})\log\frac{1}{\varepsilon}\right)$, which is the same as [26].

By leveraging the batch and mini-batch sizes, we obtain a counterpart of Corollary 3.3 as below.

**Corollary 3.6** *Set*

$$b_j \equiv 1, \quad B_j \equiv B = \min\left\{\frac{12\mathcal{H}^*}{\mu\varepsilon}, n\right\}, \quad \eta_j \equiv \eta = \frac{1}{6LB^{\frac{2}{3}}}$$

*Then it holds that*

$$\mathbb{E}C_{\text{comp}}(\varepsilon) = O\left(\left\{\left(\frac{\mathcal{H}^*}{\mu\varepsilon} \wedge n\right) + \frac{1}{\mu}\left(\frac{\mathcal{H}^*}{\mu\varepsilon} \wedge n\right)^{\frac{2}{3}}\right\}\log\frac{\Delta_f}{\varepsilon}\right).$$

Recall the results from Table 1, SCSG is $O\left(\frac{1}{\mu} + \frac{1}{(\mu\varepsilon)^{1/3}}\right)$ faster than SGD and is never worse than SVRG. When both $\mu$ and $\varepsilon$ are moderate, the acceleration of SCSG over SVRG is significant. Unlike the smooth case, we do not find any possible choice of setting that can achieve a better rate than Corollary 3.6.

## 4 Experiments

We evaluate SCSG and mini-batch SGD on the MNIST dataset with (1) a three-layer fully-connected neural network with $512$ neurons in each layer (FCN for short) and (2) a standard convolutional neural network LeNet [20] (CNN for short), which has two convolutional layers with $32$ and $64$ filters of size $5 \times 5$ respectively, followed by two fully-connected layers with output size $1024$ and $10$. Max pooling is applied after each convolutional layer. The MNIST dataset of handwritten digits has $50,000$ training examples and $10,000$ test examples. The digits have been size-normalized and centered in a fixed-size image. Each image is $28$ pixels by $28$ pixels. All experiments were carried out on an Amazon p2.xlarge node with a NVIDIA GK210 GPU with algorithms implemented in TensorFlow 1.0.

Due to the memory issues, sampling a chunk of data is costly. We avoid this by modifying the inner loop: instead of sampling mini-batches from the whole dataset, we split the batch $\mathcal{I}_j$ into $B_j/b_j$ mini-batches and run SVRG-type updates sequentially on each. Despite the theoretical advantage of setting $b_j = 1$, we consider practical settings $b_j > 1$ to take advantage of the acceleration obtained

by vectorization. We initialized parameters by TensorFlow's default Xavier uniform initializer. In all experiments below, we show the results corresponding to the best-tuned stepsizes.

We consider three algorithms: (1) SGD with a fixed batch size $B \in \{512, 1024\}$; (2) SCSG with a fixed batch size $B \in \{512, 1024\}$ and a fixed mini-batch size $b = 32$; (3) SCSG with time-varying batch sizes $B_j = \lceil j^{3/2} \wedge n \rceil$ and $b_j = \lceil B_j/32 \rceil$. To be clear, given $T$ epochs, the IFO complexity of the three algorithms are $TB$, $2TB$ and $2\sum_{j=1}^{T} B_j$, respectively. We run each algorithm with 20 passes of data. It is worth mentioning that the largest batch size in Algorithm 3 is $\lceil 275^{1.5} \rceil = 4561$, which is relatively small compared to the sample size $50000$.

We plot in Figure 1 the training and the validation loss against the IFO complexity—i.e., the number of passes of data—for fair comparison. In all cases, both versions of SCSG outperform SGD, especially in terms of training loss. SCSG with time-varying batch sizes always has the best performance and it is more stable than SCSG with a fixed batch size. For the latter, the acceleration is more significant after increasing the batch size to $1024$. Both versions of SCSG provide strong evidence that variance reduction can be achieved efficiently without evaluating the full gradient.

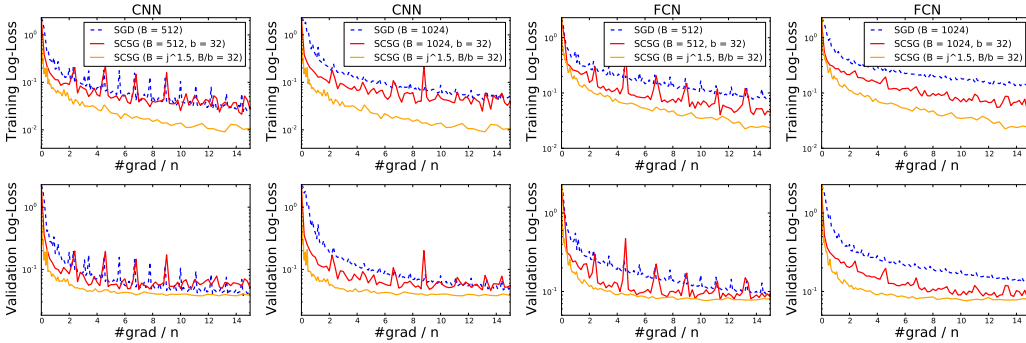

Figure 1: Comparison between two versions of SCSG and mini-batch SGD of training loss (top row) and validation loss (bottom row) against the number of IFO calls. The loss is plotted on a log-scale. Each column represents an experiment with the setup printed on the top.

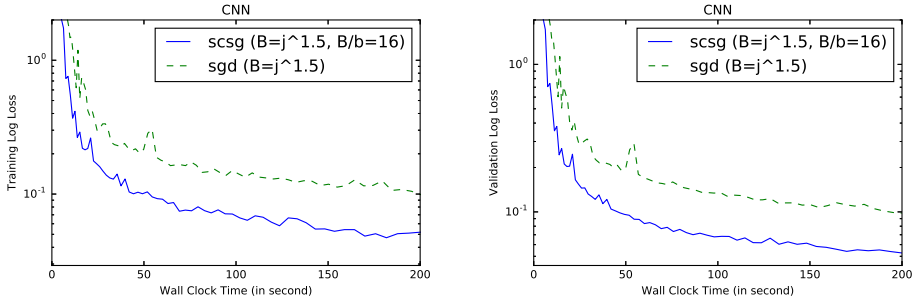

Figure 2: Comparison between SCSG and mini-batch SGD of training loss and validation loss with a CNN loss, against wall clock time. The loss is plotted on a log-scale.

Given $2B$ IFO calls, SGD implements updates on two fresh batches while SCSG replaces the second batch by a sequence of variance reduced updates. Thus, Figure 1 shows that the gain due to variance reduction is significant when the batch size is fixed. To further explore this, we compare SCSG with time-varying batch sizes to SGD with the same sequence of batch sizes. The results corresponding to the best-tuned constant stepsizes are plotted in Figure 3a. It is clear that the benefit from variance reduction is more significant when using time-varying batch sizes.

We also compare the performance of SGD with that of SCSG with time-varying batch sizes against wall clock time, when both algorithms are implemented in TensorFlow and run on a Amazon p2.xlarge node with a NVIDIA GK210 GPU. Due to the cost of computing variance reduction terms in SCSG, each update of SCSG is slower per iteration compared to SGD. However, SCSG makes faster progress

in terms of both training loss and validation loss compared to SCD in wall clock time. The results are shown in Figure 2.

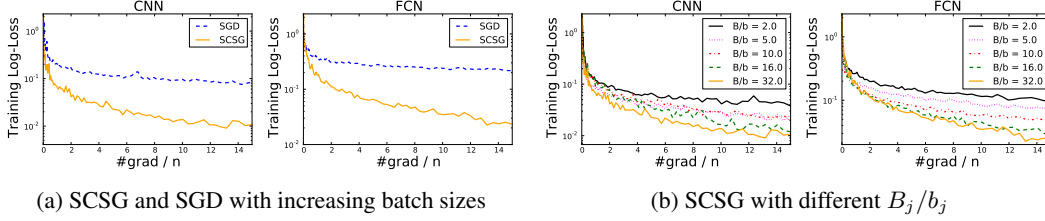

(a) SCSG and SGD with increasing batch sizes        (b) SCSG with different $B_j/b_j$

Finally, we examine the effect of $B_j/b_j$, namely the number of mini-batches within an iteration, since it affects the efficiency in practice where the computation time is not proportional to the batch size. Figure 3b shows the results for SCSG with $B_j = \lceil j^{3/2} \wedge n \rceil$ and $\lceil B_j/b_j \rceil \in \{2, 5, 10, 16, 32\}$. In general, larger $B_j/b_j$ yields better performance. It would be interesting to explore the tradeoff between computation efficiency and this ratio on different platforms.

## 5   Conclusion and Discussion

We have presented the SCSG method for smooth, non-convex, finite-sum optimization problems. SCSG is the first algorithm that achieves a uniformly better rate than SGD and is never worse than SVRG-type algorithms. When the target accuracy is low, SCSG significantly outperforms the SVRG-type algorithms. Unlike various other variants of SVRG, SCSG is clean in terms of both implementation and analysis. Empirically, SCSG outperforms SGD in the training of multi-layer neural networks.

Although we only consider the finite-sum objective in this paper, it is straightforward to extend SCSG to the general stochastic optimization problems where the objective can be written as $\mathbb{E}_{\xi \sim F} f(x; \xi)$: at the beginning of $j$-th epoch a batch of i.i.d. sample $(\xi_1, \ldots, \xi_{B_j})$ is drawn from the distribution $F$ and

$$g_j = \frac{1}{B_j} \sum_{i=1}^{B_j} \nabla f(\tilde{x}_{j-1}; \xi_i) \quad \text{(see line 3 of Algorithm 1);}$$

at the $k$-th step, a fresh sample $(\tilde{\xi}_1^{(k)}, \ldots, \tilde{\xi}_{b_j}^{(k)})$ is drawn from the distribution $F$ and

$$\nu_{k-1}^{(j)} = \frac{1}{b_j} \sum_{i=1}^{b_j} \nabla f(x_{k-1}^{(j)}; \tilde{\xi}_i^{(k)}) - \frac{1}{b_j} \sum_{i=1}^{b_j} \nabla f(x_0^{(j)}; \tilde{\xi}_i^{(k)}) + g_j \quad \text{(see line 8 of Algorithm 1).}$$

Our proof directly carries over to this case, by simply suppressing the term $I(B_j < n)$, and yields the bound $\tilde{O}(\varepsilon^{-5/3})$ for smooth non-convex objectives and the bound $\tilde{O}(\mu^{-1}\varepsilon^{-1} \wedge \mu^{-5/3}\varepsilon^{-2/3})$ for P-L objectives. These bounds are simply obtained by setting $n = \infty$ in our convergence analysis.

Compared to momentum-based methods [28] and methods with adaptive stepsizes [10, 18], the mechanism whereby SCSG achieves acceleration is qualitatively different: while momentum aims at balancing primal and dual gaps [5], adaptive stepsizes aim at balancing the scale of each coordinate, and variance reduction aims at removing the noise. We believe that an algorithm that combines these three techniques is worthy of further study, especially in the training of deep neural networks where the target accuracy is modest.

## Acknowledgments

The authors thank Zeyuan Allen-Zhu, Chi Jin, Nilesh Tripuraneni, Yi Xu, Tianbao Yang, Shenyi Zhao and anonymous reviewers for helpful discussions.

## Footnotes

[1] It is also common to use $\mathbb{E}\|\nabla f(x)\|$ to measure convergence; see, e.g. [2, 8, 9, 3]. Our results can be readily transferred to this alternative measure by using Cauchy-Schwartz inequality, $\mathbb{E}\|\nabla f(x)\| \leq \sqrt{\mathbb{E}\|\nabla f(x)\|^2}$, although not vice versa. The rates under this alternative can be made comparable to ours by replacing $\varepsilon$ by $\sqrt{\varepsilon}$.

[2]Here we allow $N$ to be zero to facilitate the analysis.

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
