[Supplementary Material]

# Supplementary Material of "Non-Convex Finite-Sum Optimization Via SCSG Methods"

**Lihua Lei**
UC Berkeley
lihua.lei@berkeley.edu

**Cheng Ju**
UC Berkeley
cju@berkeley.edu

**Jianbo Chen**
UC Berkeley
jianbochen@berkeley.edu

**Michael I. Jordan**
UC Berkeley
jordan@stat.berkeley.edu

## A    Technical Lemmas

In this section we present several technical lemmas that facilitate the proofs of our main results.

We start with a lemma on the variance of the sample mean (without replacement).

**Lemma A.1** *Let $x_1, \ldots, x_M \in \mathbb{R}^d$ be an arbitrary population of $N$ vectors with*

$$\sum_{j=1}^{M} x_j = 0.$$

*Further let $\mathcal{J}$ be a uniform random subset of $\{1, \ldots, M\}$ with size $m$. Then*

$$\mathbb{E}\left\| \frac{1}{m}\sum_{j\in\mathcal{J}} x_j \right\|^2 = \frac{M-m}{(M-1)m}\cdot\frac{1}{M}\sum_{j=1}^{M}\|x_j\|^2 \leq \frac{I(m<M)}{m}\cdot\frac{1}{M}\sum_{j=1}^{M}\|x_j\|^2.$$

**Proof**  Let $W_j = I(j \in \mathcal{J})$, then it is easy to see that

$$\mathbb{E}W_j^2 = \mathbb{E}W_j = \frac{m}{M}, \quad \mathbb{E}W_jW_{j'} = \frac{m(m-1)}{M(M-1)}. \tag{1}$$

Then the sample mean can be rewritten as

$$\frac{1}{m}\sum_{j\in\mathcal{J}} x_j = \frac{1}{m}\sum_{i=1}^{n} W_j x_j.$$

This implies that

$$\mathbb{E}\left\| \frac{1}{m}\sum_{j\in\mathcal{J}} x_j \right\|^2 = \frac{1}{m^2}\left( \sum_{j=1}^{M}\mathbb{E}W_j^2\|x_j\|^2 + \sum_{j\neq j'}\mathbb{E}W_jW_{j'}\langle x_j, x_{j'}\rangle \right)$$

$$= \frac{1}{m^2}\left( \frac{m}{M}\sum_{j=1}^{M}\|x_j\|^2 + \frac{m(m-1)}{M(M-1)}\sum_{j\neq j'}\langle x_j, x_{j'}\rangle \right)$$

$$= \frac{1}{m^2}\left(\left(\frac{m}{M} - \frac{m(m-1)}{M(M-1)}\right)\sum_{j=1}^{M}\|x_j\|^2 + \frac{m(m-1)}{M(M-1)}\left\|\sum_{j=1}^{M}x_j\right\|^2\right)$$

$$= \frac{1}{m^2}\left(\frac{m}{M} - \frac{m(m-1)}{M(M-1)}\right)\sum_{j=1}^{M}\|x_j\|^2$$

$$= \frac{M-m}{(M-1)m}\cdot\frac{1}{M}\sum_{j=1}^{M}\|x_j\|^2.$$

∎

Since the geometric random variable $N_j$ plays an important role in the analysis, we present the key property as below.

**Lemma A.2** *Let $N \sim \mathrm{Geom}(\gamma)$ for some $B > 0$. Then for any sequence $D_0, D_1, \ldots$,*

$$\mathbb{E}(D_N - D_{N+1}) = \left(\frac{1}{\gamma} - 1\right)(D_0 - \mathbb{E}D_N).$$

**Proof** By definition,

$$\mathbb{E}(D_N - D_{N+1}) = \sum_{n\geq 0}(D_n - D_{n+1})\cdot\gamma^n(1-\gamma)$$

$$= (1-\gamma)\left(D_0 - \sum_{n\geq 1}D_n(\gamma^{n-1} - \gamma^n)\right) = (1-\gamma)\left(\frac{1}{\gamma}D_0 - \sum_{n\geq 0}D_n(\gamma^{n-1} - \gamma^n)\right)$$

$$= (1-\gamma)\left(\frac{1}{\gamma}D_0 - \frac{1}{\gamma}\sum_{n\geq 0}D_n\gamma^n(1-\gamma)\right) = \left(\frac{1}{\gamma} - 1\right)(D_0 - \mathbb{E}D_N).$$

∎

**Lemma A.3** *For any $\eta > 1$ and $z > 0$, define $g_\eta(x)$ and $x(z)$ as*

$$g_\eta(x) = \frac{1 + \log x}{x^\eta}, \quad x(z) = z^{-\frac{1}{\eta}}\cdot\left(\frac{2}{\eta}\log\frac{1}{z} \vee 2\right)^{\frac{1}{\eta-1}}.$$

*Then*

$$g_\eta(x) \leq z \quad \forall x \geq x(z).$$

**Proof** For any $x \geq x(z)$, denote $\alpha = x/z^{-\frac{1}{\eta}}$. Then

$$\alpha \geq \left(\frac{2}{\eta}\log\frac{1}{z} \vee 2\right)^{\frac{1}{\eta-1}} \geq 1$$

and

$$g_\eta(x) = \frac{1 + \log\alpha + \frac{1}{\eta}\log\frac{1}{z}}{\alpha^\eta z^{-1}} \leq z\cdot\frac{2(1+\log\alpha)\left(\frac{1}{\eta}\log\frac{1}{z} \vee 1\right)}{\alpha^\eta}.$$

Taking the logarithm of both sides, we obtain that

$$\log g_\eta(x) - \log z \leq \log(1 + \log\alpha) - \eta\log\alpha + \log\left(\frac{2}{\eta}\log\frac{1}{z} \vee 2\right)$$

$$\leq \log\left(\frac{2}{\eta}\log\frac{1}{z} \vee 2\right) - (\eta - 1)\log\alpha$$

$$\leq 0.$$

∎

# B One-Epoch Analysis

As in the standard analysis of stochastic gradient methods, We start by establishing a bound of $\mathbb{E}_{\tilde{\mathcal{I}}_k}\|\nu_k^{(j)}\|^2$ and $\mathbb{E}_{\mathcal{I}_j}\|e_j\|^2$.

**Lemma B.1** *Under Assumption A1,*

$$\mathbb{E}_{\tilde{\mathcal{I}}_k}\|\nu_k^{(j)}\|^2 \leq \frac{L^2}{b_j}\|x_k^{(j)} - x_0^{(j)}\|^2 + 2\|\nabla f(x_k^{(j)})\|^2 + 2\|e_j\|^2.$$

**Proof** Using the fact that $\mathbb{E}\|Z\|^2 = \mathbb{E}\|Z - \mathbb{E}Z\|^2 + \|\mathbb{E}Z\|^2$ (for any random variable $Z$), we have

$$\mathbb{E}_{\tilde{\mathcal{I}}_k}\|\nu_k^{(j)}\|^2 = \mathbb{E}_{\tilde{\mathcal{I}}_k}\|\nu_k^{(j)} - \mathbb{E}_{\tilde{\mathcal{I}}_k}\nu_k^{(j)}\|^2 + \|\mathbb{E}_{\tilde{\mathcal{I}}_k}\nu_k^{(j)}\|^2$$

$$=\mathbb{E}_{\tilde{\mathcal{I}}_k}\|\nabla f_{\tilde{\mathcal{I}}_k}(x_k^{(j)}) - \nabla f_{\tilde{\mathcal{I}}_k}(x_0^{(j)}) - (\nabla f(x_k^{(j)}) - \nabla f(x_0^{(j)}))\|^2 + \|\nabla f(x_k^{(j)}) + e_j\|^2$$

$$\leq\mathbb{E}_{\tilde{\mathcal{I}}_k}\|\nabla f_{\tilde{\mathcal{I}}_k}(x_k^{(j)}) - \nabla f_{\tilde{\mathcal{I}}_k}(x_0^{(j)}) - (\nabla f(x_k^{(j)}) - \nabla f(x_0^{(j)}))\|^2 + 2\|\nabla f(x_k^{(j)})\|^2 + 2\|e_j\|^2.$$

By Lemma A.1,

$$\mathbb{E}_{\tilde{\mathcal{I}}_k}\|\nabla f_{\tilde{\mathcal{I}}_k}(x_k^{(j)}) - \nabla f_{\tilde{\mathcal{I}}_k}(x_0^{(j)}) - (\nabla f(x_k^{(j)}) - \nabla f(x_0^{(j)}))\|^2$$

$$\leq\frac{1}{b_j} \cdot \frac{1}{n}\sum_{i=1}^{n}\|\nabla f_i(x_k^{(j)}) - \nabla f_i(x_0^{(j)}) - (\nabla f(x_k^{(j)}) - \nabla f(x_0^{(j)}))\|^2$$

$$=\frac{1}{b_j} \cdot \left(\frac{1}{n}\sum_{i=1}^{n}\|\nabla f_i(x_k^{(j)}) - \nabla f_i(x_0^{(j)})\|^2 - \|(\nabla f(x_k^{(j)}) - \nabla f(x_0^{(j)}))\|^2\right)$$

$$\leq\frac{1}{b_j} \cdot \frac{1}{n}\sum_{i=1}^{n}\|\nabla f_i(x_k^{(j)}) - \nabla f_i(x_0^{(j)})\|^2$$

$$\leq\frac{1}{b_j} \cdot L^2\|x_k^{(j)} - x_0^{(j)}\|^2,$$

where the last line uses Assumption **A**1. Therefore,

$$\mathbb{E}_{\tilde{\mathcal{I}}_k}\|\nu_k^{(j)}\|^2 \leq \frac{L^2}{b_j}\|x_k^{(j)} - x_0^{(j)}\|^2 + 2\|\nabla f(x_k^{(j)})\|^2 + 2\|e_j\|^2.$$

■

**Lemma B.2**

$$\mathbb{E}_{\mathcal{I}_j}\|e_j\|^2 \leq \frac{I(B_j < n)}{B_j} \cdot \mathcal{H}^*.$$

**Proof** Since $\tilde{x}_{j-1}$ is independent of $\mathcal{I}_j$, conditioning on $\tilde{x}_{j-1}$ and applying Lemma A.1, we have

$$\mathbb{E}_{\mathcal{I}_j}\|e_j\|^2 = \frac{n - B_j}{(n-1)B_j} \cdot \frac{1}{n}\sum_{i=1}^{n}\|\nabla f_i(\tilde{x}_{j-1}) - \nabla f(\tilde{x}_{j-1})\|^2$$

$$\leq \frac{n - B_j}{(n-1)B_j} \cdot \mathcal{H}^* \leq \frac{I(B_j < n)}{B_j} \cdot \mathcal{H}^*$$

■

Based on Lemma A.2, Lemma B.1 and Lemma B.2, we can derive bounds for primal and dual gaps respectively.

**Lemma B.3** *Suppose $\eta_j L < 1$, then under Assumption A1,*

$$\eta_j B_j(1 - \eta_j L)\mathbb{E}\|\nabla f(\tilde{x}_j)\|^2 + \eta_j B_j\mathbb{E}\langle e_j, \nabla f(\tilde{x}_j)\rangle$$

$$\leq b_j \mathbb{E}\left(f(\tilde{x}_{j-1}) - f(\tilde{x}_j)\right) + \frac{L^3\eta_j^2 B_j}{2b_j}\mathbb{E}\|\tilde{x}_j - \tilde{x}_{j-1}\|^2 + L\eta_j^2 B_j\mathbb{E}\|e_j\|^2. \tag{2}$$

*where $\mathbb{E}$ denotes the expectation with respect to all randomness.*

**Proof** By (4) in the page 3 of the main text,

$$\mathbb{E}_{\tilde{\mathcal{I}}_k}f(x_{k+1}^{(j)}) \leq f(x_k^{(j)}) - \eta_j\langle\mathbb{E}_{\tilde{\mathcal{I}}_k}\nu_k, \nabla f(x_k^{(j)})\rangle + \frac{L\eta_j^2}{2}\mathbb{E}_{\tilde{\mathcal{I}}_k}\|\nu_k\|^2$$

$$=f(x_k^{(j)}) - \eta_j\|\nabla f(x_k^{(j)})\|^2 - \eta_j\langle e_j, \nabla f(x_k^{(j)})\rangle + \frac{L\eta_j^2}{2}\mathbb{E}_{\tilde{\mathcal{I}}_k}\|\nu_k\|^2$$

$$\leq f(x_k^{(j)}) - \eta_j(1 - \eta_j L)\|\nabla f(x_k^{(j)})\|^2 - \eta_j\langle e_j, \nabla f(x_k^{(j)})\rangle$$
$$+ \frac{L^3\eta_j^2}{2b_j}\|x_k^{(j)} - x_0^{(j)}\|^2 + L\eta_j^2\|e_j\|^2. \quad \text{(Lemma B.1)}$$

Let $\mathbb{E}_j$ denotes the expectation over $\tilde{\mathcal{I}}_0, \tilde{\mathcal{I}}_1, \ldots$, given $N_j$. Note that $\mathbb{E}_j$ is equivalent to the expectation over $\tilde{\mathcal{I}}_0, \tilde{\mathcal{I}}_1, \ldots$ as $N_j$ is independent of them. Since $\tilde{\mathcal{I}}_{k+1}, \tilde{\mathcal{I}}_{k+2}, \ldots$ are independent of $x_k^{(j)}$, the above inequality implies that

$$\eta_j(1 - \eta_j L)\mathbb{E}_j\|\nabla f(x_k^{(j)})\|^2 + \eta_j\mathbb{E}_j\langle e_j, \nabla f(x_k^{(j)})\rangle \tag{3}$$

$$\leq \mathbb{E}_j f(x_k^{(j)}) - \mathbb{E}_j f(x_{k+1}^{(j)}) + \frac{L^3\eta_j^2}{2b_j}\mathbb{E}_j\|x_k^{(j)} - x_0^{(j)}\|^2 + L\eta_j^2\|e_j\|^2. \tag{4}$$

Let $k = N_j$ in (4). By taking expectation with respect to $N_j$ and using Fubini's theorem, we arrive at

$$\eta_j(1 - \eta_j L)\mathbb{E}_{N_j}\mathbb{E}_j\|\nabla f(x_{N_j}^{(j)})\|^2 + \eta_j\mathbb{E}_{N_j}\mathbb{E}_j\langle e_j, \nabla f(x_k^{(j)})\rangle$$

$$\leq \mathbb{E}_{N_j}\left(\mathbb{E}_j f(x_{N_j}^{(j)}) - \mathbb{E}_j f(x_{N_j+1}^{(j)})\right) + \frac{L^3\eta_j^2}{2b_j}\mathbb{E}_{N_j}\mathbb{E}_j\|x_{N_j}^{(j)} - x_0^{(j)}\|^2 + L\eta_j^2\|e_j\|^2$$

$$= \frac{b_j}{B_j}\left(f(x_0^{(j)}) - \mathbb{E}_j\mathbb{E}_{N_j}f(x_{N_j}^{(j)})\right) + \frac{L^3\eta_j^2}{2b_j}\mathbb{E}_j\mathbb{E}_{N_j}\|x_{N_j}^{(j)} - x_0^{(j)}\|^2 + L\eta_j^2\|e_j\|^2. \text{ (Lemma A.2)} \tag{5}$$

The lemma is then proved by substituting $x_{N_j}^{(j)}(x_0^{(j)})$ by $\tilde{x}_j(\tilde{x}_{j-1})$, and taking expectation over all past randomness. ∎

**Lemma B.4** *Suppose $\eta_j^2 L^2 B_j < b_j^2$, then under Assumption A1,*

$$\left(b_j - \frac{\eta_j^2 L^2 B_j}{b_j}\right)\mathbb{E}\|\tilde{x}_j - \tilde{x}_{j-1}\|^2 + 2\eta_j B_j\mathbb{E}\langle e_j, \tilde{x}_j - \tilde{x}_{j-1}\rangle$$

$$\leq -2\eta_j B_j\mathbb{E}\langle\nabla f(\tilde{x}_j), \tilde{x}_j - \tilde{x}_{j-1}\rangle + 2\eta_j^2 B_j\mathbb{E}\|\nabla f(\tilde{x}_j)\|^2 + 2\eta_j^2 B_j\mathbb{E}\|e_j\|^2. \tag{6}$$

**Proof** Since $x_{k+1}^{(j)} = x_k^{(j)} - \eta_j\nu_k^{(j)}$, we have

$$\mathbb{E}_{\tilde{\mathcal{I}}_k}\|x_{k+1}^{(j)} - x_0^{(j)}\|^2$$

$$=\|x_k^{(j)} - x_0^{(j)}\|^2 - 2\eta_j\langle\mathbb{E}_{\tilde{\mathcal{I}}_k}\nu_k^{(j)}, x_k^{(j)} - x_0^{(j)}\rangle + \eta_j^2\mathbb{E}_{\tilde{\mathcal{I}}_k}\|\nu_k^{(j)}\|^2$$

$$=\|x_k^{(j)} - x_0^{(j)}\|^2 - 2\eta_j\langle\nabla f(x_k^{(j)}), x_k^{(j)} - x_0^{(j)}\rangle - 2\eta_j\langle e_j, x_k^{(j)} - x_0^{(j)}\rangle + \eta_j^2\mathbb{E}_{\tilde{\mathcal{I}}_k}\|\nu_k^{(j)}\|^2$$

$$\leq \left(1 + \frac{\eta_j^2 L^2}{b_j}\right)\|x_k^{(j)} - x_0^{(j)}\|^2 - 2\eta_j\langle\nabla f(x_k^{(j)}), x_k^{(j)} - x_0^{(j)}\rangle - 2\eta_j\langle e_j, x_k^{(j)} - x_0^{(j)}\rangle$$

$$+ 2\eta_j^2\|\nabla f(x_k^{(j)})\|^2 + 2\eta_j^2\|e_j\|^2. \quad \text{(Lemma B.1)}$$

Using the same notation $\mathbb{E}_j$ as in the proof of Lemma B.3, we have

$$2\eta_j\mathbb{E}_j\langle\nabla f(x_k^{(j)}), x_k^{(j)} - x_0^{(j)}\rangle + 2\eta_j\mathbb{E}_j\langle e_j, x_k^{(j)} - x_0^{(j)}\rangle$$

$$\leq \left(1 + \frac{\eta_j^2 L^2}{b_j}\right) \mathbb{E}_j \|x_k^{(j)} - x_0^{(j)}\|^2 - \mathbb{E}_j \|x_{k+1}^{(j)} - x_0^{(j)}\|^2 + 2\eta_j^2 \|\nabla f(x_k^{(j)})\|^2 + 2\eta_j^2 \|e_j\|^2. \quad (7)$$

Let $k = N_j$ in (7). By taking expectation with respect to $N_j$ and using Fubini's theorem, we arrive at

$$2\eta_j \mathbb{E}_{N_j} \mathbb{E}_j \langle \nabla f(x_{N_j}^{(j)}), x_{N_j}^{(j)} - x_0^{(j)} \rangle + 2\eta_j \mathbb{E}_j \langle e_j, x_{N_j}^{(j)} - x_0^{(j)} \rangle$$

$$\leq \left(1 + \frac{\eta_j^2 L^2}{b_j}\right) \mathbb{E}_{N_j} \mathbb{E}_j \|x_{N_j}^{(j)} - x_0^{(j)}\|^2 - \mathbb{E}_{N_j} \mathbb{E}_j \|x_{N_j+1}^{(j)} - x_0^{(j)}\|^2 + 2\eta_j^2 \mathbb{E}_{N_j} \|\nabla f(x_{N_j}^{(j)})\|^2 + 2\eta_j^2 \|e_j\|^2$$

$$= \left(-\frac{b_j}{B_j} + \frac{\eta_j^2 L^2}{b_j}\right) \mathbb{E}_{N_j} \mathbb{E}_j \|x_{N_j}^{(j)} - x_0^{(j)}\|^2 + 2\eta_j^2 \mathbb{E}_{N_j} \|\nabla f(x_{N_j}^{(j)})\|^2 + 2\eta_j^2 \|e_j\|^2. \text{ (Lemma A.2)}$$

$$(8)$$

The lemma is then proved by substituting $x_{N_j}^{(j)}(x_0^{(j)})$ by $\tilde{x}_j(\tilde{x}_{j-1})$ and taking expectation further on the past randomness. ∎

**Lemma B.5**
$$b_j \mathbb{E}\langle e_j, \tilde{x}_j - \tilde{x}_{j-1} \rangle = -\eta_j B_j \mathbb{E}\langle e_j, \nabla f(\tilde{x}_j) \rangle - \eta_j B_j \mathbb{E}\|e_j\|^2. \quad (9)$$

**Proof** Let $M_k^{(j)} = \langle e_j, x_k^{(j)} - x_0^{(j)} \rangle$. By definition, we have

$$\mathbb{E}_{N_j} \langle e_j, \tilde{x}_j - \tilde{x}_{j-1} \rangle = \mathbb{E}_{N_j} M_{N_j}^{(j)}.$$

Since $N_j$ is independent of $(x_0^{(j)}, e_j)$, this implies that

$$\mathbb{E}\langle e_j, \tilde{x}_j - \tilde{x}_{j-1} \rangle = \mathbb{E} M_{N_j}^{(j)}.$$

Also we have $M_0^{(j)} = 0$. On the other hand,

$$\mathbb{E}_{\tilde{\mathcal{I}}_k} \left( M_{k+1}^{(j)} - M_k^{(j)} \right) = \mathbb{E}_{\tilde{\mathcal{I}}_k} \langle e_j, x_{k+1}^{(j)} - x_k^{(j)} \rangle = -\eta_j \langle e_j, \mathbb{E}_{\tilde{\mathcal{I}}_k} \nu_k^{(j)} \rangle$$

$$= -\eta_j \langle e_j, \nabla f(x_k^{(j)}) \rangle - \eta_j \|e_j\|^2.$$

Using the same notation $\mathbb{E}_j$ as in the proof of Lemma B.3 and Lemma B.4, we have

$$\mathbb{E}_j \left( M_{k+1}^{(j)} - M_k^{(j)} \right) = -\eta_j \langle e_j, \mathbb{E}_j \nabla f(x_k^{(j)}) \rangle - \eta_j \|e_j\|^2. \quad (10)$$

Let $k = N_j$ in (10). By taking an expectation with respect to $N_j$ and using Lemma A.2, we obtain that

$$\frac{b_j}{B_j} \mathbb{E}_{N_j} M_{N_j}^{(j)} = -\eta_j \langle e_j, \mathbb{E}_{N_j} \mathbb{E}_j \nabla f(x_{N_j}^{(j)}) \rangle - \eta_j \|e_j\|^2.$$

The lemma is then proved by substituting $x_{N_j}^{(j)}(x_0^{(j)})$ by $\tilde{x}_j(\tilde{x}_{j-1})$ and taking a further expectation with respect to the past randomness. ∎

**Proof [Theorem 3.1]** Multiplying equation (2) by 2, equation (6) by $\frac{b_j}{\eta_j B_j}$ and summing them, we obtain that

$$2\eta_j B_j (1 - \eta_j L - \frac{b_j}{B_j}) \mathbb{E}\|\nabla f(\tilde{x}_j)\|^2 + \frac{b_j^3 - \eta_j^2 L^2 b_j B_j - \eta_j^3 L^3 B_j^2}{b_j \eta_j B_j} \mathbb{E}\|\tilde{x}_j - \tilde{x}_{j-1}\|^2$$

$$+ 2\eta_j B_j \mathbb{E}\langle e_j, \nabla f(\tilde{x}_j) \rangle + 2b_j \mathbb{E}\langle e_j, \tilde{x}_j - \tilde{x}_{j-1} \rangle$$

$$\leq -2b_j \mathbb{E}\langle \nabla f(\tilde{x}_j), \tilde{x}_j - \tilde{x}_{j-1} \rangle + 2b_j \mathbb{E}(f(\tilde{x}_{j-1}) - f(\tilde{x}_j)) + \left(2L\eta_j^2 B_j + 2\eta_j b_j\right) \mathbb{E}\|e_j\|^2. \quad (11)$$

By Lemma B.5, the second row can be simplified as

$$2\eta_j B_j \mathbb{E}\langle e_j, \nabla f(\tilde{x}_j) \rangle + 2b_j \mathbb{E}\langle e_j, \tilde{x}_j - \tilde{x}_{j-1} \rangle = -2\eta_j B_j \mathbb{E}\|e_j\|^2.$$

Using the fact that $2\langle a, b\rangle \le \beta\|a\|^2 + \frac{1}{\beta}\|b\|^2$ for any $\beta > 0$, we have

$$
\begin{aligned}
&- 2b_j\mathbb{E}\langle\nabla f(\tilde{x}_j), \tilde{x}_j - \tilde{x}_{j-1}\rangle \\
&\le \frac{b_j\eta_j B_j}{b_j^3 - \eta_j^2 L^2 b_j B_j - \eta_j^3 L^3 B_j^2} \cdot b_j^2\mathbb{E}\|\nabla f(\tilde{x}_j)\|^2 + \frac{b_j^3 - \eta_j^2 L^2 b_j B_j - \eta_j^3 L^3 B_j^2}{b_j\eta_j B_j}\mathbb{E}\|\tilde{x}_j - \tilde{x}_{j-1}\|^2.
\end{aligned}
$$

Putting the pieces together, we conclude that

$$
\begin{aligned}
&\frac{\eta_j B_j}{b_j}\left(2 - \frac{2b_j}{B_j} - 2\eta_j L - \frac{b_j^3}{b_j^3 - \eta_j^2 L^2 b_j B_j - \eta_j^3 L^3 B_j^2}\right)\mathbb{E}\|\nabla f(\tilde{x}_j)\|^2 \\
&\le 2\mathbb{E}(f(\tilde{x}_{j-1}) - f(\tilde{x}_j)) + \frac{2\eta_j B_j}{b_j}\left(1 + \eta_j L + \frac{b_j}{B_j}\right)\mathbb{E}\|e_j\|^2. \qquad (12)
\end{aligned}
$$

Since $\eta_j L = \theta_j = \gamma(b_j/B_j)^{\frac{2}{3}}$ and $b_j \ge 1, B_j \ge 8b_j \ge 8$,

$$
b_j^3 - \eta_j^2 L^2 b_j B_j - \eta_j^3 L^3 B_j^2 = b_j^3\left(1 - \gamma^2 \cdot b_j^{-\frac{2}{3}} B_j^{-\frac{1}{3}} - \gamma^3 \cdot b_j^{-1}\right) \ge b_j^3(1 - \gamma^2/2 - \gamma^3).
$$

Then (12) can be simplified as

$$
\begin{aligned}
&\gamma\left(\frac{B_j}{b_j}\right)^{\frac{1}{3}}\left(2 - \frac{2b_j}{B_j} - 2\gamma\left(\frac{b_j}{B_j}\right)^{\frac{2}{3}} - \frac{1}{1 - \gamma^2/2 - \gamma^3}\right)\mathbb{E}\|\nabla f(\tilde{x}_j)\|^2 \\
&\le 2L\mathbb{E}(f(\tilde{x}_{j-1}) - f(\tilde{x}_j)) + 2\gamma\left(1 + \gamma\left(\frac{b_j}{B_j}\right)^{\frac{2}{3}} + \frac{b_j}{B_j}\right)\left(\frac{B_j}{b_j}\right)^{\frac{1}{3}}\mathbb{E}\|e_j\|^2 \\
&\le 2L\mathbb{E}(f(\tilde{x}_{j-1}) - f(\tilde{x}_j)) + 2\gamma\left(1 + \gamma\left(\frac{b_j}{B_j}\right)^{\frac{2}{3}} + \frac{b_j}{B_j}\right)\frac{I(B_j < n)}{b_j^{\frac{1}{3}} B_j^{\frac{2}{3}}} \cdot \mathcal{H}^* \text{ (By Lemma B.2).}
\end{aligned}
$$
$$(13)$$

Since $B_j \ge 8b_j, \gamma \le \frac{1}{3}$, we have

$$
2 - \frac{2b_j}{B_j} - 2\gamma\left(\frac{b_j}{B_j}\right)^{\frac{2}{3}} - \frac{1}{1 - \gamma^2/2 - \gamma^3} \ge 2 - \frac{1}{4} - \frac{\gamma}{2} - \frac{1}{1 - \gamma^2/2 - \gamma^3} \ge 0.482
$$

and

$$
1 + \gamma\left(\frac{b_j}{B_j}\right)^{\frac{2}{3}} + \frac{b_j}{B_j} \le 1 + \frac{\gamma}{4} + \frac{1}{8} \le 1.209.
$$

Thus, (13) implies that

$$
\left(\frac{B_j}{b_j}\right)^{\frac{1}{3}}\mathbb{E}\|\nabla f(\tilde{x}_j)\|^2 \le \frac{5L}{\gamma} \cdot \mathbb{E}(f(\tilde{x}_{j-1}) - f(\tilde{x}_j)) + \frac{6I(B_j < n)}{b_j^{\frac{1}{3}} B_j^{\frac{2}{3}}} \cdot \mathcal{H}^*. \qquad (14)
$$

$\blacksquare$

## C   Convergence Analysis for Smooth Objectives

**Proof [Theorem 3.2]** Since $\tilde{x}_T^*$ is a random element from $(\tilde{x}_j)_{j=1}^T$ with

$$
P(\tilde{x}_T^* = \tilde{x}_j) \propto \frac{\eta_j B_j}{b_j} \propto (B_j/b_j)^{\frac{1}{3}},
$$

we have

$$
\mathbb{E}\|\nabla f(\tilde{x}_T^*)\|^2 \le \frac{\frac{5L}{\gamma} \cdot \mathbb{E}(f(\tilde{x}_0) - f(\tilde{x}_{T+1})) + 6\left(\sum_{j=1}^T b_j^{-\frac{1}{3}} B_j^{-\frac{2}{3}} I(B_j < n)\right)\mathcal{H}^*}{\sum_{j=1}^T b_j^{-\frac{1}{3}} B_j^{\frac{1}{3}}}
$$

$$\leq \frac{\frac{5L}{\gamma} \cdot (f(\tilde{x}_0) - f^*) + 6\left(\sum_{j=1}^{T} b_j^{-\frac{1}{3}} B_j^{-\frac{2}{3}} I(B_j < n)\right) \mathcal{H}^*}{\sum_{j=1}^{T} b_j^{-\frac{1}{3}} B_j^{\frac{1}{3}}}.$$

∎

**Proof** [**Corollary 3.3**] By Theorem 3.2,

$$\mathbb{E}\|\nabla f(\tilde{x}_T^*)\|^2 \leq \frac{30\Delta_f + 6TB^{-\frac{2}{3}} I(B < n) \cdot \mathcal{H}^*}{TB^{\frac{1}{3}}} = \frac{30\Delta_f}{TB^{\frac{1}{3}}} + \frac{6\mathcal{H}^* \cdot I(B < n)}{B}.$$

Let $\tilde{T}(\epsilon)$ be the minimum number of epochs such that

$$\frac{30\Delta_f}{\tilde{T}(\epsilon)B^{\frac{1}{3}}} \leq \frac{\epsilon}{2}.$$

Then under the setting of the corollary, for any $T \geq \tilde{T}(\epsilon)$,

$$\mathbb{E}\|\nabla f(\tilde{x}_T^*)\|^2 \leq \frac{\epsilon}{2} + \frac{6\mathcal{H}^* \cdot I(B < n)}{B} \leq \frac{\epsilon}{2} \leq \epsilon.$$

By definition, we know that $T(\epsilon) \leq \tilde{T}(\epsilon)$. Noticing that

$$\tilde{T}(\epsilon) = O\left(\left\lceil \frac{\Delta_f}{\epsilon B^{\frac{1}{3}}} \right\rceil\right) = O\left(1 + \frac{\Delta_f}{\epsilon B^{\frac{1}{3}}}\right),$$

we conclude that

$$\mathbb{E}C_{\text{comp}}(\epsilon) = O(T(\epsilon)B) = O(\tilde{T}(\epsilon)B) = O\left(B + \frac{\Delta_f}{\epsilon} \cdot B^{\frac{2}{3}}\right).$$

The corollary is then proved by substituting for $B$. ∎

**Proof** [**Corollary 3.4**] By Theorem 3.2,

$$\mathbb{E}\|\nabla f(\tilde{x}_T^*)\|^2 \leq \frac{30\Delta_f + 6\sum_{j=1}^{T} \frac{I(B_j<n)}{j}\mathcal{H}^*}{\sum_{j=1}^{T}(j^{\frac{1}{2}} \wedge n^{\frac{1}{3}})} \triangleq W(T).$$

Let $T_* = \lfloor n^{\frac{2}{3}} \rfloor$. First we prove that $W(T)$ is strictly decreasing.

1. When $T \geq T_*$, the numerator is a constant and the denominator is strictly increasing. Thus, $W(T)$ is strictly decreasing on $[T_*, \infty)$;

2. When $T < T_*$, let $a_{1j} = \frac{6\mathcal{H}^*}{j}$ and $a_{2j} = j^{\frac{1}{2}}$. Further let

$$U(T) = \frac{30\Delta_f}{\sum_{j=1}^{T} a_{2j}}, \quad V(T) = \frac{\sum_{j=1}^{T} a_{1j}}{\sum_{j=1}^{T} a_{2j}},$$

then

$$W(T) = U(T) + V(T).$$

It is obvious that $U(T)$ is strictly decreasing. Noticing that $\frac{a_{1j}}{a_{2j}} = \frac{6\mathcal{H}^*}{j^{\frac{3}{2}}}$ is strictly decreasing, we also conclude that $V(T)$ is stricly decreasing. Therefore, $W(T)$ is strictly decreasing on $[1, T_*]$.

In summary, $W(T)$ is stricly decreasing. Now we show that for any $T \geq T(\epsilon)$,

$$W(T) \leq \epsilon,$$

which implies that $\mathbb{E}\|\nabla f(\tilde{x}_T)\|^2 \leq \epsilon$.

To do so, we distinguish two cases to analyze $W(T)$.

1. If $T \leq T_*$, then
$$W(T) = \frac{30\Delta_f + 6\left(\sum_{j=1}^{T} \frac{1}{j}\right)\mathcal{H}^*}{\sum_{j=1}^{T} j^{\frac{1}{2}}}.$$

Since $\frac{1}{j}$ is decreasing, we have
$$\sum_{j=1}^{T} \frac{1}{j} = 1 + \sum_{j=2}^{T} \frac{1}{j} \leq 1 + \int_{1}^{T} \frac{dx}{x} = 1 + \log T.$$

Similarly, since $j^{\frac{1}{2}}$ is increasing, we have
$$\sum_{j=1}^{T} j^{\frac{1}{2}} \geq \int_{0}^{T} x^{\frac{1}{2}} dx = \frac{2}{3} T^{\frac{3}{2}}.$$

Therefore,
$$W(T) \leq \frac{45\Delta_f + 9(1 + \log T)\mathcal{H}^*}{T^{\frac{3}{2}}}.$$

2. If $T > T_*$, then
$$W(T) = \frac{30\Delta_f + 6\left(\sum_{j=1}^{T_*} \frac{1}{j}\right)\mathcal{H}^*}{\sum_{j=1}^{T_*} j^{\frac{1}{2}} + n^{\frac{1}{3}}(T - T_*)} = W(T_*) \cdot \frac{\sum_{j=1}^{T_*} j^{\frac{1}{2}}}{\sum_{j=1}^{T_*} j^{\frac{1}{2}} + n^{\frac{1}{3}}(T - T_*)}. \tag{15}$$

Similar to the first case, we have
$$\sum_{j=1}^{T_*} j^{\frac{1}{2}} = \sum_{j=1}^{T_*-1} j^{\frac{1}{2}} + T_*^{\frac{1}{2}} \leq \int_{1}^{T_*} \sqrt{x}\,dx + n^{\frac{1}{3}} = \frac{2}{3}n + n^{\frac{1}{3}} - \frac{2}{3}.$$

Since
$$n^{\frac{1}{3}} - \frac{2}{3} = \frac{n-1}{n^{\frac{2}{3}} + n^{\frac{1}{3}} + 1} + \frac{1}{3} \leq \frac{n}{3},$$

we obtain that
$$\sum_{j=1}^{T_*} j^{\frac{1}{2}} \leq n.$$

As a result, (15) implies that
$$W(T) \leq W(T_*) \cdot \frac{1}{1 + n^{-\frac{2}{3}}(T - T_*)}$$

Putting the pieces together, we obtain that
$$W(T) \leq \begin{cases} \dfrac{45\Delta_f + 9(1 + \log T)\mathcal{H}^*}{T^{\frac{3}{2}}} & \triangleq W_1(T) \quad (T \leq T_*) \\ \dfrac{W(T_*)}{1 + n^{-\frac{2}{3}}(T - T_*)} & \triangleq W_2(T) \quad (T > T_*) \end{cases}. \tag{16}$$

It is easy to see that both $W_1(T)$ and $W_2(T)$ are strictly decreasing and $\lim_{T \to \infty} W_1(T) = \lim_{T \to \infty} W_2(T) = 0$. Let
$$T_1(\epsilon) = \min\{T : W_1(T) \leq \epsilon\}, \quad T_2(\epsilon) = \min\{T \geq T_* : W_2(T) \leq \epsilon\}.$$

Recall that $W(T)$ is also strictly decreasing, we have
$$T(\epsilon) \leq \begin{cases} T_1(\epsilon) & (W(T_*) \leq \epsilon) \\ T_2(\epsilon) & (W(T_*) > \epsilon) \end{cases}.$$

More concisely,
$$T(\epsilon) \leq T_1(\epsilon) \wedge T_* + (T_2(\epsilon) - T_*). \tag{17}$$

To derive a bound for $T_1(\epsilon)$, let $\tilde{T}_1(\epsilon)$ be the minimum $T$ such that

$$\frac{45\Delta_f}{T^{\frac{3}{2}}} \leq \frac{\epsilon}{2}, \quad \frac{9\mathcal{H}^*(1+\log T)}{T^{\frac{3}{2}}} \leq \frac{\epsilon}{2}.$$

Then by Lemma A.3, we have

$$T_1(\epsilon) \leq \tilde{T}_1(\epsilon) = O\left(\left(\frac{\Delta_f}{\epsilon}\right)^{\frac{2}{3}} + \left(\frac{\mathcal{H}^*}{\epsilon}\right)^{\frac{2}{3}} \cdot \log^2\left(\frac{\mathcal{H}^*}{\epsilon}\right)\right).$$

On the other hand, it is straightforward to derive a bound for $T_2(\epsilon)$ as

$$T_2(\epsilon) - T_* \leq \left(n^{\frac{2}{3}} \cdot \frac{W(T_*) - \epsilon}{\epsilon}\right)_+ \leq n^{\frac{2}{3}} \cdot \frac{W(T_*)}{\epsilon} = O\left(\frac{\Delta_f}{\epsilon n^{\frac{1}{3}}} + \frac{\mathcal{H}^* \log n}{\epsilon n^{\frac{1}{3}}}\right).$$

Therefore, we conclude that

$$T(\epsilon) = O\left(\min\left\{\frac{1}{\epsilon^{\frac{2}{3}}}\left[\Delta_f^{\frac{2}{3}} + (\mathcal{H}^*)^{\frac{2}{3}} \log^2\left(\frac{\mathcal{H}^*}{\epsilon}\right)\right], n^{\frac{2}{3}}\right\} + \frac{\Delta_f + \mathcal{H}^* \log n}{\epsilon n^{\frac{1}{3}}}\right). \qquad (18)$$

Finally, to obtain the bound for the computation complexity, we notice that

$$\sum_{j=1}^{T} j^{\frac{3}{2}} = O(T^{\frac{5}{2}}).$$

Let $x_+$ denote the positive part of $x$, i.e., $x_+ = \max\{x, 0\}$. Therefore,

$$\mathbb{E}C_{\text{comp}}(\epsilon) = O\left(\sum_{j=1}^{T(\epsilon)} B_j\right) = O\left((T(\epsilon) \wedge T_*)^{\frac{5}{2}} + n(T(\epsilon) - T_*)_+\right)$$

$$= O\left((T_1(\epsilon) \wedge T_*)^{\frac{5}{2}} + n(T_2(\epsilon) - T_*)\right)$$

$$= O\left(\min\left\{\frac{1}{\epsilon^{\frac{5}{3}}}\left[\Delta_f^{\frac{5}{3}} + (\mathcal{H}^*)^{\frac{5}{3}} \log^5\left(\frac{\mathcal{H}^*}{\epsilon}\right)\right], n^{\frac{5}{3}}\right\} + \frac{n^{\frac{2}{3}}}{\epsilon} \cdot (\Delta_f + \mathcal{H}^* \log n)\right).$$

$\blacksquare$

**Remark 1** *The log-factor* $\log^5\left(\frac{1}{\epsilon}\right)$ *can be reduced to* $\log^{\frac{3}{2}+\mu}\left(\frac{1}{\epsilon}\right)$ *for any* $\mu > 0$ *by setting* $B_j = \lceil j^{\frac{3}{2}}(\log j)^{\frac{3}{2}+\mu} \wedge n\rceil$. *In this case,*

$$W(T) = \frac{30\Delta_f + 6\left(\sum_{j=1}^{T} \frac{I(B_j < n)}{j(\log j)^{1+\frac{2\mu}{3}}}\right)\mathcal{H}^*}{\sum_{j=1}^{T} j^{\frac{1}{2}}(\log j)^{\frac{1}{2}+\frac{\mu}{3}}}.$$

*For any* $\mu > 0$,

$$\sum_{j=1}^{T} \frac{I(B_j < n)}{j(\log j)^{1+\frac{2\mu}{3}}} \leq 1 + \int_1^{\infty} \frac{1}{x(\log x)^{1+\frac{2\mu}{3}}} < \infty.$$

*On the other hand, as proved above,*

$$\sum_{j=1}^{T} j^{\frac{1}{2}}(\log j)^{\frac{1}{2}+\frac{\mu}{3}} \geq \sum_{j=1}^{T} j^{\frac{1}{2}} \sim T^{\frac{3}{2}}.$$

*Thus,*

$$W(T) \sim O\left(\frac{\Delta_f + \mathcal{H}^*}{T^{\frac{3}{2}}}\right).$$

*Using similar arguments and treating* $\Delta_f, \mathcal{H}^* = O(1)$ *for simplicity, we can obtain that*

$$T(\epsilon) = O\left(\epsilon^{-\frac{2}{3}} \wedge n^{\frac{2}{3}} + \frac{1}{\epsilon n^{\frac{1}{3}}}\right).$$

*If $B_{T(\epsilon)} < n$, then*

$$\mathbb{E}C_{\text{comp}}(\epsilon) = O\left(\sum_{j=1}^{T(\epsilon)} j^{\frac{3}{2}}(\log j)^{\frac{3}{2}+\mu}\right) = O\left(T(\epsilon)^{\frac{5}{2}} \cdot (\log T(\epsilon))^{\frac{3}{2}+\mu}\right) = O\left(\epsilon^{-\frac{5}{3}}\log^{\frac{3}{2}+\mu}\left(\frac{1}{\epsilon}\right)\right).$$

*If $B_{T(\epsilon)} \geq n$, we obtain the same bound as in Corollary 3.4.*

## D  Convergence Analysis for P-L Objectives

**Proof** [**Theorem 3.5**] By equation (14) in the proof of Theorem 3.2 (see p.6) and the P-L condition,

$$\mu\left(\frac{B_j}{b_j}\right)^{\frac{1}{3}}\mathbb{E}(f(\tilde{x}_j) - f^*) \leq \left(\frac{B_j}{b_j}\right)^{\frac{1}{3}}\mathbb{E}\|\nabla f(\tilde{x}_j)\|^2$$

$$\leq \frac{5L}{\gamma} \cdot \mathbb{E}(f(\tilde{x}_{j-1}) - f(\tilde{x}_j)) + 6b_j^{-\frac{1}{3}}B_j^{-\frac{2}{3}}I(B_j < n) \cdot \mathcal{H}^*.$$

For brevity, we write $F_j$ for $\mathbb{E}(f(\tilde{x}_j) - f^*)$. Then

$$\left(\mu\gamma B_j^{\frac{1}{3}} + 5Lb_j^{\frac{1}{3}}\right)F_j \leq 5Lb_j^{\frac{1}{3}}F_{j-1} + 6\gamma B_j^{-\frac{2}{3}}I(B_j < n) \cdot \mathcal{H}^*. \qquad (19)$$

By definition of $\lambda_j$, this can be reformulated as

$$F_j \leq \lambda_j F_{j-1} + 6\gamma\mathcal{H}^* \cdot \frac{I(B_j < n)}{\mu\gamma B_j + 5Lb_j^{\frac{1}{3}}B_j^{\frac{2}{3}}}.$$

Apply the above inequality iteratively for $j = T, T-1, \ldots, 1$, we prove the result. ∎

**Proof** [**Corollary 3.6**] When $B_j \equiv B, b_j \equiv 1$ and $\gamma = \frac{1}{6}$, (19) in the proof of Theorem 3.5 can be reformulated as

$$\left(\mu\gamma B^{\frac{1}{3}} + 30L\right)\left(F_j - \frac{6\mathcal{H}^*I(B < n)}{\mu B}\right) \leq 30L\left(F_{j-1} - \frac{6\mathcal{H}^*I(B < n)}{\mu B}\right).$$

This implies that

$$F_T \leq \left(\frac{30L}{\mu B^{\frac{1}{3}} + 30L}\right)^T \Delta_f + \frac{6\mathcal{H}^*I(B < n)}{\mu B}.$$

Under the setting of this problem,

$$\frac{6\mathcal{H}^*I(B < n)}{\mu B} \leq \frac{\epsilon}{2}.$$

By definition of $T(\epsilon)$, we have

$$T(\epsilon) \leq \log\frac{\Delta_f}{\epsilon} \bigg/ \log\left(\frac{30L}{\mu B^{\frac{1}{3}} + 30L}\right) = O\left(\log\frac{\Delta_f}{\epsilon}\left(1 + \frac{L}{\mu B^{\frac{1}{3}}}\right)\right).$$

As a consequence,

$$\mathbb{E}C_{\text{comp}}(\epsilon) = O\left(T(\epsilon)B\right) = O\left(\left(B + \frac{LB^{\frac{2}{3}}}{\mu}\right)\log\frac{\Delta_f}{\epsilon}\right).$$

Plugging into $B$, we end up with

$$\mathbb{E}C_{\text{comp}}(\epsilon) = O\left(\left\{\left(\frac{\mathcal{H}^*}{\mu\epsilon} \wedge n\right) + \frac{1}{\mu}\left(\frac{\mathcal{H}^*}{\mu\epsilon} \wedge n\right)^{\frac{2}{3}}\right\}\log\frac{\Delta_f}{\epsilon}\right)$$

∎