[Reviews · NeurIPS 2017]

Reviewer 1



Summary: This paper proposes a variant of a family of stochastic optimization algorithms SCSG (closely related to SVRG), and analyzes it in the context of non-convex optimization. The main difference is that the outer loop computes a gradient on a random subset of the data, and the stochastic gradients in the inner loop have a random cardinality and are not restricted to those participating in the outer gradient. The analysis of the stochastic gradients despite this mismatch is the main technical contribution of the paper. The result is a set of new bounds on the performance of the algorithm which improve on both sgd and variance reduced algorithms, at least in the low-medium accuracy regimes. Experimental evidence supports the claims of improvement in objective reduction (both training error and test error) per iteration both before the end of the first pass on data and after, both by SCSG over SGD and by varying batch size variant over the fixed size variant, but only on two networks applied to the venerable and small MNIST dataset. Pros of acceptance: - The claimed results are interesting and shed light on an important regime. - The proof method might be more widely applicable. Cons: - The experimental section is insufficient for the claims made. Quality: Theoretical contribution: The main contribution of the paper seems to be an analysis of the discrepancy due to using partial long-term gradients, using the behavior of sampled means without replacement. If novel, this is a useful contribution to our understanding of a trick that is practically important for SVRG-type algorithms. I’ve spot checked a few proofs in the supplementary material, and found a minor error: - Lemma B.3 assumes eta_j*L < 1, but table 2 does not mention L in giving the step sizes. The empirical evaluation uses only one small and old dataset, which is bad methodology (there are many old small datasets, any reasonable algorithm will be good at one). Other than that, it is well designed, and the comparison in terms of number of gradients is reasonable, but should be complemented by a wall-time comparison as some algorithms have more overhead than others. The paper claims SCSG is never worse than SVRG and better in early stages; this justifies a direct comparison to SVRG (with B = n) for each part. An added plot in the existing comparison for early stages, and also a separate experiment that compares (on a log-log plot) the rates of local convergence to high precision inside a basin of attraction. Clarity: The paper is written quite well: the algorithm is clear including the comparison to previous variants of SCSG and of SVRG, the summary of bounds is helpful, and the presentation of bounds in appropriate detail in different parts. Some minor comments: - L.28 table 1: o What is “best achievable” referring to in [26,27]? A lower bound? An algorithm proven optimal in some sense? should that be "best available" instead? - L.30 - 33: these sentences justify the comparison to SGD in a precision regime, but are not very clear. Substituting at the desired regime (eps = n^-1/2 or some such) would make the bounds directly comparable. - L.56 - 59 are confusing. Applying variance reduction in the non-convex is not novel. - Algorithm 1, L.2: \subset instead of \in, for consistency use the concise notation of 7. - L.106: “In order to minimize the amount the tuning works” something is wrong with that sentence. - L.116: “and the most of the novelty” first “the” is probably superfluous. Supplementary material: - L.8 n should be M - L.13 B > 0 does not appear in the rest of the lemma or in the proof. Presume it is \gamma? - Originality: I leave to reviewers more up to date with the non-convex literature to clarify whether the claimed novelties are such. Significance: This paper seems to complete the analysis of the existing SCSG, a practical SVRG variant, to the common non-convex regime. Specifically, showing theoretically that it improves on SGD in early stages is quite important. ----- Updates made after rebuttal phase: - The authors addressed the issues I'd raised with the proofs. One was a typo, one was a misinterpretation on my part. There may well still be quality issues, but they appear to be minor. I removed the corresponding con. I think the paper could be an 8 if its empirical evaluation was not so flawed.

Reviewer 2



Very interesting result. From theoretical perspective the paper is strong, well organized and self-sufficient. The only comment that I have is experimental evaluation. MNIST data set is not the best choice for classification problems with Neural Networks. I strongly recommend to try ImageNet. (http://www.cv-foundation.org/openaccess/content_cvpr_2016/html/He_Deep_Residual_Learning_CVPR_2016_paper.html)

Reviewer 3



The authors propose a stochastically controlled stochastic gradient method for smooth non-convex optimization. They prove a convergence rate for such method that is superior to the usual SGD and GD methods in terms of the computational complexity. The theoretical results are interesting and relatively novel. But in order to show the benefits of the algorithm, much more extensive experiments are needed than the ones presented in the paper. The biggest concern regarding the experiments in the paper is that they do not include the comparison to other accelerating methods, e.g. momentum based. The authors merely compare their method to SGD and not to any other method. In my opinion, although the other acceleration techniques are different with the proposed method, it is worth understanding how much benefit each one of them brings to the table and at what cost. It is interesting that the authors have presented experimental results with respect to the number of gradient evaluations as a surrogate for time. But as the authors it is mentioned in the paper, due to the possibility of vectorized computation the gradient computation might not be a suitable surrogate for time. Therefore, I recommend presenting some experiments with wall clock time as a reference for comparison in addition to the current ones with #gradient evaluations. Although using time as a reference can depend on the implementation and platform, but at the end of the day, it is what is important in practice.